# Artificial Intelligence and Machine Learning in Prostate Cancer Patient Management—Current Trends and Future Perspectives

**DOI:** 10.3390/diagnostics11020354

**Published:** 2021-02-20

**Authors:** Octavian Sabin Tătaru, Mihai Dorin Vartolomei, Jens J. Rassweiler, Oșan Virgil, Giuseppe Lucarelli, Francesco Porpiglia, Daniele Amparore, Matteo Manfredi, Giuseppe Carrieri, Ugo Falagario, Daniela Terracciano, Ottavio de Cobelli, Gian Maria Busetto, Francesco Del Giudice, Matteo Ferro

**Affiliations:** 1The Institution Organizing University Doctoral Studies (I.O.S.U.D.), George Emil Palade University of Medicine, Pharmacy, Sciences and Technology from Târgu Mureș, 540142 Târgu Mureș, Romania; sabin.tataru@gmail.com (O.S.T.); osan.virgil@gmail.com (O.V.); 2Department of Cell and Molecular Biology, George Emil Palade University of Medicine, Pharmacy, Sciences and Technology from Târgu Mureș, 540142 Târgu Mureș, Romania; 3Department of Urology, Medical University of Vienna, 1090 Vienna, Austria; 4Department of Urology, SLK Kliniken Heilbronn, University of Heidelberg, 74074 Heilbronn, Germany; jens.rassweiler@slk-kliniken.de; 5Department of Emergency and Organ Transplantation-Urology, Andrology and Kidney Transplantation Unit, University of Bari, 70124 Bari, Italy; giuseppe.lucarelli@inwind.it; 6Department of Urology, San Luigi Gonzaga Hospital, University of Turin, Orbassano, 10143 Turin, Italy; porpiglia@libero.it (F.P.); daniele.amparore@unito.it (D.A.); matteo.manfredi85@gmail.com (M.M.); 7Department of Urology and Organ Transplantation, University of Foggia, 71122 Foggia, Italy; giuseppe.caririeri@unifg.it (G.C.); ugofalagario@gmail.com (U.F.); 8Department of Translational Medical Sciences, University of Naples Federico II, 80131 Naples, Italy; daniela.terracciano@unina.it; 9Division of Urology, European Institute of Oncology (IEO)-IRCCS, 20141 Milan, Italy; ottavio.decobelli@ieo.it (O.d.C.); matteo.ferro@ieo.it (M.F.); 10Department of Oncology and Haematology-Oncology, Università degli Studi di Milano, 20122 Milan, Italy; 11Department of Urology and Renal Transplantation, University of Foggia, Policlinico Riuniti of Foggia, 71122 Foggia, Italy; gianmaria.busetto@unifg.it; 12Department of Urology, Policlinico Umberto I, Sapienza University of Rome, 00185 Rome, Italy; francesco.delgiudice@uniroma1.it

**Keywords:** prostate cancer, biomarker, genomics, artificial intelligence, artificial neural network

## Abstract

Artificial intelligence (AI) is the field of computer science that aims to build smart devices performing tasks that currently require human intelligence. Through machine learning (ML), the deep learning (DL) model is teaching computers to learn by example, something that human beings are doing naturally. AI is revolutionizing healthcare. Digital pathology is becoming highly assisted by AI to help researchers in analyzing larger data sets and providing faster and more accurate diagnoses of prostate cancer lesions. When applied to diagnostic imaging, AI has shown excellent accuracy in the detection of prostate lesions as well as in the prediction of patient outcomes in terms of survival and treatment response. The enormous quantity of data coming from the prostate tumor genome requires fast, reliable and accurate computing power provided by machine learning algorithms. Radiotherapy is an essential part of the treatment of prostate cancer and it is often difficult to predict its toxicity for the patients. Artificial intelligence could have a future potential role in predicting how a patient will react to the therapy side effects. These technologies could provide doctors with better insights on how to plan radiotherapy treatment. The extension of the capabilities of surgical robots for more autonomous tasks will allow them to use information from the surgical field, recognize issues and implement the proper actions without the need for human intervention.

## 1. Introduction

Prostate cancer (PCa) is the third most commonly diagnosed cancer worldwide, after lung and breast cancer and the fifth cause of cancer-specific death in males [1]. Around 191,930 patients will be diagnosed with PCa in 2020 in the United States, with an estimated 33,330 deaths [2]. In the last few years, research was focused on diagnosis, prognosis and prediction of PCa outcomes taking a leap through the use of Statistics and Artificial Intelligence (AI). The use of computer-based learning models has become a predominant area of research in PCa.

Artificial Neural Networks (ANN) have increasingly been used to build advanced prognostic models for PCa [3]. To train a machine learning model it is enough to acquire structured datasets including input variables and outcomes, with little knowledge of the PCa insights. For instance, several novel tools are available for screening and diagnosis of PCa such as genomics, magnetic resonance imaging (MRI) and biomarkers (exosomes and molecular imaging). In this scenario, AI may have a pivotal role, first in the interpretation of this enormous amount of data, second in the development of machine learning algorithms that may help urologists to reduce the number of unnecessary prostate biopsies without missing the diagnosis of aggressive PCa. Moreover, the use of genomics, AI and extracellular vesicles [4] (exosomes and cell-free DNA from body fluids), can provide a more reliable and rapid PCa test [5].

AI is defined as the ability of a computer to perceive the surrounding environment and make the same decisions as human intellect on an action, all this to reach a certain goal [6]. Machine learning (ML) is a subfield of AI and implies the creation and deployment of algorithms to analyze data and its properties and is not given a task specifically based on certain predefined inputs from the environment. ML techniques can mainly be classified according to the type of label and feature. For labeling, ML can be classified into three models such as supervised, unsupervised and reinforcement learning. For features, ML can be classified into handcrafted or non-handcrafted feature-based techniques [7]. Deep learning (DL) is a form of ML that enables machine devices (such as computers) to learn from experience and understand the environment in terms of a hierarchy of concepts. Computers gather experience in learning and a human does not need to pre-specify all the data to the computer [8]. Lately, deep convolutional neural networks (DCNNs), a modified type of AAN, have been proven to have high efficiency when applied to digitized images, a form of computer aided diagnosis (CAD) analysis. DCNNs allowed the automatic extraction of imaging features from digitized images in PCa [9] and it is now used to classify PCa and benign tissues with magnetic resonance imaging (MRI) [10]. In the last few years, DL has application in image classification, object detection, segmentation [11], the detection of anatomical and cellular structures, tissue segmentation, device aid on disease diagnosis and prognosis [12], with a new model emerging, which seems to perform better, called the massive-training artificial neural network [13]. Clinical decision support systems (CDSSs) are being developed to provide improvement in decision making. At this moment, reviews and systematic reviews give limited data on how ML and DL techniques could provide clinical application based on CDSSs in PCa oncological care [14,15]. Therefore, AI and ML are still being considered an area of development. A brief explanation and glossary of technical terms is provided in Table 1.

This review focuses on the analysis of the last five years’ published papers on AI and ML techniques by scrutinizing PubMed, Web of Science and Science Direct, using prostate cancer, biomarker, genomics, artificial intelligence and artificial neural networks keywords. We had also included the AI leading studies from later years or data on other diseases. The goal was to give an overview of the current evidence and future directions of AI in all the clinical management of prostate cancer patients from the diagnosis to the treatment.

## 2. AI in Digital Pathology of Prostate Cancer

### 2.1. Developments in Optical Image Analysis

Since 2010, Pantanowitz et al. [20] have talked about digital imaging and digital imaging processing. Digital whole histopathological slides are more interactive, easy to share, involve less preparation time and generate teaching sets (virtual slide boxes). Telepathology and image analysis by ML, using several algorithms such as computer assisted image analysis is something that has greater precision than the traditional microscope. The whole slide imaging, such as slide preservation over time, new handling of images, telepathology, quality assurance, education and collaborative research make the ML more easily understandable [21]. With the help of an augmented optical light microscope that enables the real-time integration of AI, Chen et al. [22] developed and evaluated deep learning algorithms for the fast identification of PCa. Further studies are needed to optimize the process.

### 2.2. Differentiation of Cellular Structures and Tissues

Nevertheless, quantitative morphometric features coming from image analysis may play a pivotal role in the diagnosis of PCa [23]. Some of the features have been included in automatic classifiers to differentiate stroma, normal glands and malignant tissue in PCa. With access to large sets of digitized tissue images from tissue microarrays (TMA), at the National Institutes of Health, Kwak et al. [24] used five different methods to identify cellular structures and the performance of the multi-view boosting methods. For the differentiation of cancer from benign tissue, the multi-view boosting classification showed a significantly higher AUC (area under the curve) (0.98 95% CI 0.97–0.99) compared to the single-view classifications (*p*-value < 0.01) and outperformed the single view approach, which will increase the accuracy, robustness and utility of digital pathology tools analysis of tissues. In an observational study, Arvaniti et al. [25], on a training dataset of TMAs from 641 patients, small image patches extracted from benign tissue and PCa annotated regions were used to train a patch-based classifier. The stratification achieved by the authors’ deep learning patch model, separated the low-risk and intermediate-risk groups significantly greater than the one achieved by either pathologist (Benjamini-Hochberg-BH-corrected two-sample log rank *p*-value  =  0.098), respectively (BH-corrected two-sample log rank *p*-values  =  0.79 and 0.292) and the model revealed a close inter-observer agreement (with kappa = 0.71 and 0.75) which was seen between two ground truth pathologists (kappa = 0.71). 

### 2.3. Demographic and Histopathology Reports Data 

Roffman et al. [26] developed and validated a multiparametric ANN for PCa risk prediction and stratification. Based on clinical and demographic characteristics, the pre-histopathological status allowed the model to predict PCa risk. The ANN model yielded high specificity (89.4%) and low sensitivity (23.2%) for the prediction of prostate cancer risk. Lenain et al. [27], enlisting histopathological data from 4470 PCa patients, used different machine learning approaches to analyze the staging information for T(tumor), N(nodes) and M(metastasis), such as support vector machines (SVM), random forest (RF), and extreme gradient boosting (XGB). They used for analyzing the ML model performance model, the precision, recall and F-Score a measure of a model’s accuracy on a dataset [28], as the metrics and obtained classification of pathology best result for N (F1-Score 0.98) and M (F1-Score 0.99).

### 2.4. Whole Slide Image Analysis in Prostate Biopsies (Cancer Detection and Grading) and Performance Comparison of AI Models and Pathologists

Ström et al. [29] digitized 6682 slides from needle core biopsies of 976 patients and 271 from 93 men outside the study, then trained a DNN, evaluating the prediction of its presence, its extent and Gleason grade of cancer. The correlation between model and assigned pathologist was AUC 0.96 to detect cancer and a mean pair wise kappa 0.62 for assigning Gleason grades. Litjens et al. [30], with the data from 225 glass slides of PCa biopsies, trained a deep learning CNN, a network that can be applied to every pixel in a whole slide image, this in order to detect PCa in the biopsy specimens, with an AUC for the 90th percentile analysis of 0.98 in slide detection cancer. Campanella et al. [31] evaluated the framework of whole slide images for prostate, with the goal of avoiding expensive and time-consuming pixel-wise manual annotations, obtaining best results on the prostate dataset (N = 24,859) with an AUC of 0.989 at 20 times optical magnification and the automated removal of 75% of slides resulted in no loss in sensitivity. One year before, Campanella et al. [32], using 12,160 whole slide images, 2424 positive and 9736 negative, trained on the full dataset an AlexNet and a ResNet18 network, and pretrained on various image models on ImageNet, achieved results with their best model on the ResNet34 and VGG11-BN of 0.976 and 0.977 AUC, respectively. In PCa classification from histopathologic images, patch-wise cross-validation and single pathologist lead to biases, therefore using patient based cross-validation and the opinion of several experts [33,34], because it is well known that the PCa grading, which is variable with the experience of urologic pathologist [35], will lead to the better performance of classification methods. Nagpal et al. [36], trained a DL system on 1557 slides, and compared it to a reference standard provided by 29 pathology experts (mean accuracy was 0.61 on the validation dataset), and the model reached a higher diagnostic accuracy of 0.70 (*p* = 0.002), and is better leaning for patient stratification risk. A convolutional network analysis (CNN) by Lucas et al. [37] shows that, with proper training, the CNN can differentiate areas that are not atypical and malignant areas with an accuracy of 92%, with a sensitivity and specificity of 90% and 93%, respectively. Lately, Raciti et al. [38] published the results of how an AI system like Paige Prostate Alpha can influence pathologists during the diagnosis of PCa on biopsy needle cores. In diagnosing PCa with Paige Prostate Alpha, sensitivity for all pathologists increased with the AI system (average sensitivity without Paige Prostate Alpha: 74% ± 11%; with Paige Prostate Alpha: 90% ± 4%). To determine the Gleason grade groups, they found an increase in average sensitivity with a Paige Prostate Alpha of 20% for Grade group 1, 13% for Grade group 2 and 11% for Grade group 3, allowing the pathologist to better classify lower grade groups.

A population-based, diagnostic study aimed to identify PCa by an AI system, training deep neural network (DNN) can lead to better identification of the malignant from the benign cores of prostate biopsies (AUC 0·997(95% CI 0.994–0.999) for the independent test dataset (benign = 910, malignant = 721) and 0·986 (0.972–0.996) for the external validation dataset (benign = 108, malignant = 222) and further fine tuning of the AI system will allow for better stratifying Gleason grading groups because they identified that the performance dropped in cancer length predictions and overall Gleason grading [29]. 

In another large trial published by Bulten et al. [39], they analyzed 5759 PCa core biopsies from 1243 patients and trained the deep-learning system to detect Gleason grade. In the test dataset, the deep-learning system achieved an AUC of 0·990 on determining the malignancy of a biopsy, on the observer set an AUC of 0.984 and the system outperformed 10 out of 15 pathologists and achieved the same results as the reference standard both for Grade group 2 or more (AUC 0.978, 0.966–0.988), and a grade group of 3 or more (AUC 0.974, 0.962–0.984). There are also limitations to the system because pathologists can assess the volume more qualitatively and the system counts the exact area of the individual glands and the findings of Egevad et al. [40] should be used to improve calibration of AI systems. Therefore, Bulten et al. [41] published yet other results showing that the limitations can be addressed if AI is assisting pathologists rather than having a competition in assessing the performance of either pathologists or AI systems. The list of studies discussing different types of algorithms used to train the models, which were looking into Gleason grading of biopsies, TMAs, whole section, the amount of data used and their results are listed in Table 2.

In summary, all evidence up to this moment shows that the AI systems are in need of further development to help and assist pathologists to provide an accurate diagnosis. AI could provide in the future the necessary aid in histopathological diagnosis, especially in remote areas and health systems that need the expertise of highly trained pathologists. Before any clinical setting usage of such systems they will have to be approved by health regulators. The quest for personalized approach in PCa, the quantitative histopathological diagnosis will have to provide pathologist with new tools to increase the sensitivity and specificity of more accurate readings of the tissue images and AI seems to provide this better and faster.

## 3. AI in Diagnostic Imaging of Prostate Cancer

### 3.1. ML in MRI Imaging Tools in PCa 

Early attempts to fusion pathology with imaging have been performed since 1991 by Schnall et al. [42], with limited success to correlate radiographic and pathological features. In 2012, Ward et al. [43] evaluated a technique to perform digital registration of images from histopathology and in vivo MRI using image-guided specimen slicing based on strand-shaped fiducial markers. Since 2014, Litjens et al. [44] have used segmentation and characterization of MRI images using several models (analyzing digital prostate images, pixels for basic image analysis) represents the base for the more recently discovered and applied methods. Segmentation of prostate is very important for identifying its deformable capsule, has application in prostate fusion biopsy, brachytherapy and can be done with the help of MRI and transrectal ultrasonography (TRUS). In a multi-institutional study, Gaur et al. [45] have shown that artificial intelligence based detection, improved specificity combined with PI-RADS (Prostate Imaging-Reporting and Data system) v.2 (version 2) categorization and they found a sensitivity for the described images for PIRADSv.2 ≥ 3 of 78%. Here, the greatest benefit was seen in the transitional zone (TZ), where it helped moderately experienced readers to achieve the level of the well experienced radiologists and help to improve the reading of the transitional zone was of 83.8% sensitivity with automated detection versus 66.9% with MRI alone [45].

Abdollahi et al. [46] used radiomics based models on T2 weighted images (T2W) that were more reliable than apparent diffusing coefficient (ADC) in MRI data, as apparent diffusing coefficient for Gleason score for staging of the disease seems to perform better. Lately, Dulhanty et al. [47] have used novel radiomics with ADC and computed high-b value diffusion weighted imaging (CHB-DWI) modalities for prostate cancer diagnosis with a better performance than a clinical heuristics driven strategy. Another DL based approach, presented by Aldoj et al. [48], a CNN using different 3D combinations (ADC, diffusion weighted imaging (DWI), T2 weighted images) with an AUC of 0.91 at 81.2% sensitivity and 90.5% specificity, compared to a radiologist using PI-RADS v2 [49].

The two PROSTATEx Challenges were an effort to improve, with the help of CAD, the classification of clinically significant PCa and the characterization of the Gleason grade group. The PROSTATEx Challenge involved quantitative image analysis methods to analyze prostatic lesions, and the PROSTATEx-2 involved quantitative MRI biomarkers for the determination of Gleason grade group in PCa [50]. As time passed, in 2020, de Vente et al. [51] characterized another computerized model, involving deep learning regression, in bi-parametric MRI examination, soft-label ordinal regression improves the performance of PCa grading and detection from biparametric-MRI over earlier presented methods. A retrospective analysis [52], using the data collected in 2017 from the PROSTATEx Challenges, trained a CNN model and the best input achieved in this study was a combination of T2-weighted images, ADC and DWI, and the results were that an end-to-end training of the CNN model, with data from different scanners and protocols, can be generalized and more validation in the future is needed. To minimize the interference of different MRI scanners and image acquisition protocols, Sunoqrot et al. [53] proposed a model that uses automatic (dual fat and muscle reference approach), signal intensity normalization to improve T2-weighted MR images of the prostate using object recognition, and significantly higher AUC (0.826 vs. 0.769) for the classification of histologically diagnosed peripheral zone tissues compared to the other methods. Because of the variation between diagnoses of different radiologist using PIRADS for assessing prostate lesions, a DL was developed further to help with the characterization of clinically significant PCa, and the results were that the CNN trained has the similar power as an experienced radiologist [54]. New MRI techniques are developed to improve the quality and acquisition time of image sequences, quantitative imaging, computer-aided diagnosis and artificial intelligence (luminal water imaging -LWI, Restriction spectrum imaging–RSI, vascular, extracellular and restricted diffusion for cytometry in tumors-VERDICT, hybrid multi-dimensional MRI-(HM-MRI), MR fingerprinting (MRF), segmented or multi-shot DWI), to better detect, characterize, diagnose and predict prognoses for PCa [55]. Many of the examples of algorithms used have to train to define each zone that is in a spatial label on MRI, to be in accordance with the Gleason score of each tumor. For sure, spatial annotation at full resolution of the digital histopathological images will improve the ability of the machine learning techniques to differentiate normal versus malignant tissues [56].

### 3.2. ML in TRUS Imaging in PCa

Karimi et al. [33] identified a model that means patch-based training and evaluation could lead to significant overestimation of a model’s predictive accuracy. The importance is that patient-based training and evaluation is the only acceptable method for developing machine learning models in this application. The DL network is trained to extract and to learn its own features on the basis of the raw image to improve the classification of the image compared with the ML approach [7]. 

AI can help by decreasing the time of reader that interprets the data; therefore, there will be an increase in the performance of radiologists. Applications of AI to prostate MRI can specifically increase the sensitivity of PCa detection and decrease inter-reader variability [57]. 

Compared to the clinical TRUS segmentation, the Zeng at al. [58] method that used the CNN statistical shape model was able to achieve lower volume matching results in all prostate regions, but especially at the base and apex. Karimi et al. [59] proposed a new CNN architecture for prostate clinical target volume segmentation in TRUS images, which computes multi-scale features directly from the input images, which learns similarities in all training images, then cross-validated them, and developed a method to improve uncertain segmentations based on the estimated uncertainty map and the expected shape. Feng et al. [60] proposed a new method, which extracts features from both the spatial and the temporal dimensions by performing three-dimensional convolution operations and when compared to other methods this method achieved a sensitivity of 82.98 ± 6.23, a specificity of 91.45 ± 6.75 and an accuracy of 90.18 ± 6.62 in PCa detection using contrast enhanced ultrasonography (CEUS), anti-PSMA (prostate specific membrane antigen) and the non-targeted blank agent as contrast agents. Wildeboer et al. [61] assessed the potential of ML B-mode, shear-wave elastography (SWE) and dynamic contrast-enhanced ultrasound (DCE-US) with a high result compared to contrast velocity with an AUC of 0.75 and 0.90 for PCa and Gleason > 3 + 4. 

### 3.3. Combined MRI and TRUS for ML in Prostate Cancer

Fusion biopsy relies on distant past frames and newly acquired in real time TRUS and proper segmentation using MRI and TRUS. Moving from state-of-the art techniques that use the deep CNN in visual recognition tasks and that have superior performance in obtaining reliable images for prostate mapping in brachytherapy techniques [62], Anas et al. [63] proposed an automatic prostate segmentation technique that incorporates temporal information of TRUS images to improve the segmentation accuracy and has the ability to enable real-time deformable registration and improved biopsy guidance. A CNN was proposed for the registration of T2-weighted MRI and 3D TRUS volumes of the prostate [64] and then a hybrid 3D/2D U-Net CNN (the Hybrid 3D/2D U-Net was trained on 3D images and then completed object detection and segmentation on 2D images) approach to prostate organ segmentation was described as having good performance in regard to prostate segmentation and volumetric evaluation [65]. Liu et al. [66] recruited 50 confirmed PCa patients with a Prostate Imaging-Reporting and Data System version 2 (PI-RADS v2) score of 4 or 5. Logistic regression based on the first and strongest enhancement phase (Dataset-FS) integration; enhanced phases of dynamic contrast enhanced magnetic resonance imaging (DCE-MRI) can lead to the diagnostic ability of the model to predict PCa invasiveness with a noninvasive accuracy of 0.90. Ishioka et al.’s [67] model of computer-aided diagnosis, based on a CNN algorithm combined with U-net with ResNet50, showed AUC values in the two evaluation data sets, 0.645 and, respectively, 0.636, for estimating the designated area in which targeted biopsy confirmed the presence of PCa, and AUC values had improved as the DL progressed and they found an improvement in diagnostic accuracy by lowering the number of patients mistakenly diagnosed as having cancer. The accuracy of an imaging diagnosis and the improvements related to developments in DL techniques could further lower the inter and intra-observer variability and make faster learning curves in reading PCa scoring systems such as PI-RADS. A summary of the included studies and their different imaging used modalities, combinations and techniques, artificial intelligence used methods and their performance are listed in Table 3.

## 4. AI in Prostate Cancer Genomics

In order to predict individual outcomes in patients with PCa, there is an increased interest in the genomics of PCa and how the alterations in the PCa genome can change the individual evolution of his PCa [68]. In the last five years there has been a scarcity of scientific data in terms of genetic research, with an increase in the last few years. 

### 4.1. ML Approach in mRNA, miRNA and SNPs (Single Nucleotide Polymorphisms)

A few data are available from 2016, when Bertoli et al. [69], using a meta-analysis approach, identified through an ML approach a group of 29 miRNAs that can be used for diagnostic purposes and a group of 7 miRNAs that may have prognostic abilities. MacInnis et al. used a novel analysis method, back in 2016, with the dependency of association on the number of Top Hits, that identified 14 regions associated with PCa using a conventional logistic regression analysis of individual single-nucleotide polymorphisms [70]. Decipher uses a random forest algorithm to predict PCa metastatic disease [7,71]. Another study of Lee et al. [72] used ML (pre-conditioned random forest regression) and bioinformatics tools to assess, on genome based study, the conditions that appear after radiotherapy and to predict late toxicity, resulting in a statistically significant prediction model (*p* = 0.01), only for weak stream. DNA methylation markers have been identified as having diagnostic ability. 

### 4.2. ML in Gene Expression and Gene Activity

Hou et al. [73] used a genetic algorithm optimized artificial neural network to establish a diagnostic model that showed good results for the diagnosis (AUC = 0.953) and prognosis (AUC of 5 years overall survival time = 0.808) of PCa. Liu et al. [74] identified 12 CpG (cytosine and guanine on a genome) site markers and 13 promoter markers, using a deep neural network model, from an initial pool of 139,422 CpG sites, and the promoter methylation data contained 15,316 promoters and applied three machine learning strategies (moderated t-statistics, LASSO, and random-forest).This might further be used for liquid biopsy of cancers. Lately, Hamzeh et al. [75] used a combination of efficient machine learning methods (Support Vector Machine (SVM)-Radial basis function kernel (SVM-RBF), Naive Bayes, Random Forest) to analyze gene activity and to identify the genes for the presence of PCa (on one side or both sides). The highest accuracy and precision for the different classifiers came from the SVM-RBF classifier, which was able to separate the different locations by an accuracy of 99% and found genes (HLA-DMB and EIF4G2) that are correlated with PCa progression. de la Calle et al. [76] aimed to predict the recurrence and progression of PCa based on biomarker analysis from 648 samples (424 tumors, 224 normal tissue) using tissue micro assays anti Ki-67, anti ERG (erythroblast transformation-specific related gene) antibodies through an AI algorithm, having 100% identification of ERG positive tumors. Genomics is playing an important role in the outcome of PCa patients. Genes could be identified through ML as candidate biomarkers or with a potential diagnostic role but for the future, great computational power will be required to increase the receiver operating characteristics in prognostic for the individual patient, through fused data streams. At this point, the scarcity of studies to report on fused data streams is linked to the challenges with the study of genomics. Further studies linked to data fused streams are required to identify the best ML methods, or to improve the existing ones, to tackle the challenges in clinical application of AI in genomics. A summarization of the discussed studies are imbedded in Table 4.

## 5. AI in Prostate Cancer Treatment

### 5.1. AI in Prostate Cancer Radiotherapy

#### 5.1.1. MRI Based ML Approach for Treatment of PCa

The artificial intelligence and machine learning techniques described for imaging different types of cancers can be extended to treatment planning that involves radiotherapy. For brachytherapy and external beam radiation therapy (EBRT), radiomics-based detection of cancerous patches described in MRI were transferred on to a computed tomography (CT) scan for EBRT, using a texture feature enabled machine learning classifier, to achieve a deformable map to accurately predict the cancer lesions [77]. For treatment planning, using a deep attention U-Net network that integrates attention gates and deep supervision, Dong et al. [84] compared models with or without deep attention algorithms. Compared to CT, deep attention networks a synthetic MRI (sMRI), especially developed for soft tissues, which obtained better results in volume overlapping, better surface matching and better center and volume matching, probably offering better PCa radiotherapy treatment planning. Savenje et al. [85] investigated the feasibility of the clinical use of organs at risk of auto-segmentation based on CNN DeepMedic and V-net, using MRI images and the qualitative analysis showed that delineation from DeepMedic required fewer adaptations and less time for the delineation procedure, therefore it is important for the optimization of clinical workflow. In an effort to investigate the accuracy of dose calculations in PCa radiotherapy, Shafai-Erfani et al. [86] used a CNN algorithm, random forest, from synthetic CT images generated from MRI images. Quantitative results showed no significant differences in dose volume histogram and planning target volumes, showing that in the future ML-MRI methods could generate synthetic CT images from MRI and could probably eliminate CT acquisition and eliminate its radiation toxicity.

#### 5.1.2. ML Approach in TRUS Studies for Treatment of PCa

For brachytherapy, ultrasound studies on prostate segmentation, a deeply supervised deep learning-based approach [78], an efficient learning-based multi-label segmentation algorithm [79] and an estimation of model uncertainty and use of prior shape information to significantly improve the performance of CNN-based medical image segmentation methods [59], show clinical feasibility in terms of accuracy compared to manual segmentation of the prostate and potential benefit for clinical applications that can help clinicians for training and decision support for intraoperative planning or targeted biopsy. This includes from evaluating ML algorithms for treatment planning in prostate brachytherapy with expected improvement in prostate low dose rate treatment plans to lower planning time and resources [80], to the use of quantitative susceptibility mapping and unsupervised machine learning to accurate and robust manner to localize the radioactive seeds [87], to the implementation of a sliding-window convolutional neural network for radioactive seed identification in MRI-assisted radiosurgery. This is to improve seed identification [81] and the radiation dose calculations using CNN models to help improve the speed of those calculations [88,89,90]. The outcomes of toxicity following radiotherapy were assessed by Isaksson et al. [91] in a review of PCa radiotherapy treatment in terms of genitourinary and gastrointestinal toxicity, and the publications screened only a few that showed better performance than classical models. By adding more features when training the model (the use of statin drugs and PSA (Prostate Specific Antigen) level prior to intensity, modulated radiotherapy was found to be strongly related to the toxicity outcome. DL based methods will be able to calculate the radio-therapeutic dose with accuracy and efficiency and this will have an important role for real time radiotherapy. In the future, DL will be a valuable asset in many different aspects for both patients and clinicians. A summary of studies can be found in Table 4.

### 5.2. AI in Prostate Cancer Surgery

Robotic assisted platforms for prostate surgery (e.g., radical prostatectomy) already use artificial intelligence through machine learning based methods [92]. Some of the potential applications of ML in surgical robotics are automation of the surgical operation, saving the best strategies of high volume surgeons, training surgeons, classification and standardization of surgical procedures, safe interaction between environment and surgical robots and safe interaction between surgeons and surgical robots [93]. If we look into the future, automatizing the surgical procedure is a real desire. The complexity of an autonomous system that can perform surgical interventions is very high and there is a need to cover all aspects of a surgical procedure and to develop a system that can transfer the surgical skills toward automated execution. Sarikaya et al. [82] proposed in 2017 an end-to-end deep learning approach for instrument detection and localization in robotic assisted surgery images. By using a CNN processing stream and multimodal convolutional network, they demonstrated that the proposed model is better than similar approaches, but the process is still slow in computing all images and could represent a basis for further studies. The research on how open radical prostatectomy and robotic assisted radical prostatectomy influences the emotions of patients in the two treatment groups was assessed with Patient Reported Information Multidimensional Exploration version2, from the online discussions of patients in different support groups to automated identification and intelligent analysis of emotions [94]. The biochemical recurrence was analyzed in patients following robotic radical assisted prostatectomy using three supervised ML algorithms and multiple training variables, with the result that ML techniques can produce accurate disease predictability in all three models, which is better than traditional statistical regression [95]. Hung at al. [83] used a da Vinci system recorder to collect automated performance data, adding them to ML algorithms, and found that bimanual dexterity is an ideal surgical skill and camera manipulation strongly correlates with surgeon expertise and good outcomes. Goldenberg at al. [7] developed a system that uses a computer controlled TRUS transducer in the rectum and tracks the surgical instrument tips and the real time MRI and TRUS images. These images enable the visualization of suspected lesions in real time. AI uses software for imaging in real time, intraoperative modifications and this will require a team effort between regulatory authorities and the research and development of manufacturers of the equipment to better implement the technology in the future, to provide better clinical outcomes for the patients. A list of analyzed papers is integrated in Table 4.

## 6. Limitations and Future Perspectives

The evidence at this time, highlighted in the present papers, points out that AI could advance pathology, imaging, genomics and surgery through how we understand AI and its advantages. Now AI is still developing and advancing. The time to train an AI system is high and it is not possible without human intervention. Some of the tasks of AI can surprisingly match, in some perspectives, the performance of experts but still limitations and challenges remain [41]. Regulation is mandatory for every test that will be introduced in clinical practice. The latest regulatory papers in the European Union and the United States of America will undergo certification beyond self-validation and certification studies to prove the reproducible result of the test and to avoid its nonreproducible risk [96,97].

One of the challenges is how these AI ML models will join clinical practice. How the leading researchers will put into practice a model that can predict PCa, diagnosing it with histopathological or with imaging methods, remains a challenge [98].

It is estimated that in the future there will be commercially available tools to predict PCa, or to grade PCa with the aid of AI. But presently, an AI method is limited by small data sets, and for the models to have a broader impact, they will have to have very large and representative data sets and images. Fine tuning a DL method, standardizing and controlling the process to improve the model, would further reduce errors through the generalization of a process. Some of the limitations that will have to be surpassed are related to the costs of digitalizing images, software and hardware acquisition, the need to show pathologists that AI is safe and can be applied to large cohorts and to set a threshold at which an AI model is performing at least as well as the pathologist, which is a multi-view approach to identifying cancerous tissue and differentiating it from benign tissue. Deep learning methods seem to be the most appropriate models to be applied in histopathology, especially through image data sets analysis and classification (amount of slides, pixels, image digitalization). For the future it will be very interesting to see whether, in image slide analysis of morphometric features, images will correlate with radiological methods and proteomics. For sure, future studies will show the extent of this possible association. DCNN or DNN is the state of the art method that had achieved great accuracy in terms of classification in medical imaging [29]. But present studies cannot recommend AI in pathology in a clinical setting. Algorithms used for radiological imaging are focused on registration, segmentation and radiomics, and the trend to automatization is on. The actual methods that were studied had shown little advancements on how to deliver the information to the physician. Most of the papers presented are focused on segmentation, acquisition of images, the quality of picture obtained and in the future, transfer adaptation techniques and supervision will be of great importance [99]. Currently, algorithms analyze the lesions and require manual training of the models. The current methods try to color code the lesions and transparency to low probabilities and are the most advanced in automatic visualization [100]. The targeted biopsy still poses a challenge in accuracy. Movements, prostate image deformation and poor alignment lead to errors. An integrated system that allows real-time imaging and biopsy will limit the underdiagnoses [101]. AI and deep learning techniques will have the ability to change the inter- and intra-observer variability and the limitations given by the interpretation of scoring systems, such as PIRADS, from less experienced physicians [99]. Further research is mandatory to obtain better prediction of malignancies in PCa images, to accurately biopsy those lesions and to properly diagnose clinically significant PCa.

The diagnosis and prognosis of PCa has been guided by PSA as a biomarker. In the last ten years there have been numerous biomarkers identified with potential use in clinical practice. Those biomarkers’ potential for diagnoses and the prediction of prognosis will have to be integrated in a clinical setting. An ANN can play an important role in analyzing biomarkers such as KI-67 and ERG antibodies [14,76]. There is the potential that AI will provide fast and possibly more reliable identification and validation of biomarkers in PCa. 

Radiotherapy and AI is based on the capacity of systems to provide better MRI and TRUS images with proper segmentation and shaping of organ boundaries in order to provide the therapeutic dose for prostate, in both EBRT and brachytherapy. There are studies that used ML to develop methods to better localize radioactive seeds [81,87], using CNNs to calculate the right dose. DL based methods will be able to calculate the radio-therapeutic dose with accuracy and efficiency in order to reduce toxicity [91]. AI techniques will need future studies to better identify anatomical regions for radiotherapy, better radioactive seed implantation to cover the lesions, and dose calculations to reduce radiotherapy related toxicity.

AI software is becoming more attractive, especially for ML algorithms in the construction of 3D models, which could be integrated in augmented reality and virtual reality systems for surgical purposes. Still, obtaining new technologies to perform automatic intraoperative image overlapping is an area of development [92]. An ML that will predict surgical movements, along with visualization in real time of the organ localization of tumors, is a field that needs further research and it could be of great use to surgeons in the future.

Overall, the AI methods trained to provide assistance to clinicians have to have clinical applications. They will have to be adapted to be easily understood and used by physicians. They will have to perform at least as well the experts in their fields of pathology, imaging, radiotherapy and surgery to be accepted by clinicians as of benefit to their patients. They will to be approved by regulatory agencies across the world and this will not be an easy task. There is a need for further studies to validate AI ML methods for clinical use.

## 7. Conclusions

In PCa, artificial intelligence and ANN algorithms (especially DCNNs) are promising for diagnosis and playing a predictive role in the prognoses of the disease. With sparse evidence, the need for further studies is real. The potential for diagnostic imaging, histopathology, genomics and treatment can hold great promise for the future and can improve the individualization of the disease and therefore improve patient outcomes in a more personalized fashion. The potential of AI in prostate cancer surgery will boost training and surgery performance for the future, both in terms of improvement outcomes for the patients, and also for the benefit of training and assessing surgical skills.

## Figures and Tables

**Table 1 diagnostics-11-00354-t001:** A glossary of Artificial Intelligence (AI) Terms.

Technical Term	Short Definition
Artificial Intelligence (AI)	Any technique that enables machines to mimic human behavior [6]
Artificial Neural Network (AAN)	A mathematical statistical model imitating the human brain in processing data and creating patterns used in a decision making process [16]
Machine Learning (ML)	A subset of AI which learns from experience without being explicitly programmed in order to deliver specific outputs [17]
Deep Learning (DL)	A subset of ML structured similar to the human brain processing, using large multiple data sets at the same time, evaluating and reprocessing multiple time to reach an output [18]
Convolutional Neural Network (CNN)	An AAN that is particularly efficient when applied to digitized images and pattern recognition [19]

**Table 2 diagnostics-11-00354-t002:** AI studies, type of training in accordance to Gleason grading of biopsies, tissue microarrays (TMAs) and whole section and their results.

Article/Reference	Image Type	Image Analysis Method	Number of Slides or Patients (N)	Task	Results
Campanella et al. [31]	Whole slide images	Multiple instances learning based, deep learning	N = 24,859 slides	Automated cancer detection	Model achieved AUC over 0.98
Campanella et al. [32]	Whole slide images	Multiple instances learning based, deep learning	N = 12,160 slides	Automated cancer detection	Modelachieved an AUC of 0.98 and a false negative rate of 4.8%
Litjens et al. [30]	Whole slide images	Deep learning, CNN	N = 225 slides	Automated cancer detection	AUC for the 90th percentile analysis of 0.98 in slide detection cancer
Arvaniti et al. [25]	Tissue microarrays	Deep learning, CNN	N = 641 patients	Automated Gleason grading	Model revealed a close inter-observer agreement (with kappa = 0.71 and 0.75) which was seen between two ground truth pathologists (kappa = 0.71).
Lenain et al. [27]	Words analization	Natural language processing	N = 4470 patients	Classify free-text pathology reports	Classification of pathology best result for N (F1-Score 0.98) and M (F1-Score 0.99)
Ström et al. [29]	Whole slide images	Deep learning, CNN	N = 6682 slides	Automated cancer detection and Gleason grading	Correlation between model and assigned pathologist was AUC 0.96 to detect cancer and a mean pair wise kappa 0.62 for assigning Gleason grades
Nagpal et al. [36]	Whole slide images	Deep learning, CNN	N = 1557 slides	Automated Gleason grading	Model achieved accuracy of 0.70 compared to 29 pathologists (0.61) on the validation set, *p* = 0.002
Raciti et al. [38]	Whole slide images	Paige Prostate Alpha Deep learning, CNN	N = 304 slides	Automated cancer detection and Gleason grading	Diagnosed PCa with Paige Prostate Alpha, sensitivity for all pathologists increased with the AI system (average sensitivity without Paige Prostate Alpha: 74% ± 11%; with Paige Prostate Alpha: 90% ± 4%). An increase in average sensitivity with Paige Prostate Alpha, of 20% for Grade group 1, 13% for Grade group 2 and 11% for Grade group 3
Bulten et al. [39]	Whole slide images	Deep learning, CNN	N = 1243 patients	Automated Gleason grading	Model achieved the same results as the reference standard both for Grade group 2 or more (AUC 0.978, 0.966–0.988), and grade group of 3 or more (AUC 0.974, 0.962–0.984)

Abbreviations: AUC—area under the curve, *Abbreviations:* CNN—convolutional neural network, kappa—Cohen’s quadratic kappa statistic, AI—artificial intelligence, PCa—prostate cancer.

**Table 3 diagnostics-11-00354-t003:** Different imaging modalities and techniques, AI method and performance and combination of imaging modalities.

Article/Reference	Imaging Modality	Imaging Technique	AI Method	Performance
Litjens et al. [44]	MRI	T2W, PD, DWI, DCE	RFC	AUC 0.89
Gaur et al. [45]	MRI	T2W, DWI, DCE, ADC	RFC	Benefit in TZ for moderately-experienced readers at PI-RADSv2 < 3 (84% vs mpMRI-alone 67%, *p* = 0.055), and PI-RADSv2 ≥ 3, CAD improved patient-level specificity (72%) compared to mpMRI-alone (45%, *p* < 0.001).
Zeng et al. [58]	MRI, TRUS	T2W, B-mode	CNN	Reduction in the segmentation error at base and apex, 5–10% reduction in aRVD.
Karimi et al. [59]	TRUS	B-mode	CNN	CNN method outperformed other methods in DSC, HD *p* = 0.01
Feng et al. [60]	TRUS	CEUS	CNN	Sensitivity of 82.98 ± 6.23, specificity of 91.45 ± 6.75 and accuracy of 90.18 ± 6.62 compared to other methods
Wildeboer et al. [61]	TRUS	B-mode, SWE, CEUS	RFC	AUC 0.90 for PCa and Gleason >3 + 4, outperforming contrast velocity
Hu et al. [64]	MRI, TRUS	T2W, 3D B-mode	GMM	Median target registration error of 3.6 mm on landmark centroids and a median Dice of 0.87 on prostate glands
Liu et al. [66]	MRI	T1W, T2W, DWI, DCE	SVM, RF, RFC, DT, KNN	KNN—best predict efficacy AUC-0.88; accuracy-0.85
Ishioka et al. [67]	MRI	T2W	CNN	AUC 0.645 estimating the designated area in which targeted biopsy confirmed the presence of PCa
Abdollahi et al. [46]	MRI	T2W, ADC	CNN	For GS prediction, T2 W radiomic models more predictive (mean AUC 0.739) than ADC models (mean AUC 0.70). For stage prediction, ADC models higher prediction performance (mean AUC 0.675)
Aldoj et al. [48]	MRI	T2W, ADC, DWI	CNN	AUC of 0.91 at 81.2% sensitivity and 90.5% specificity, compared to radiologist using PI-RADS v2
de Vente et al. [51]	MRI	T2W, ADC, DWI	CNN	Voxel-wise weighted kappa of 0.446 ± 0.082 and a Dice similarity coefficient for segmenting clinically significant cancer of 0.370 ± 0.046, above ProstateX-2
Sunoqrot et al. [53]	MRI	T2W	CNN	Healthy vs. malignant classification also improved significantly (*p* < 0.001) in peripheral (AUC 0.826 vs. 0.769) and transition (AUC 0.743 vs. 0.678) zones.

Abbreviations: RFC—random forest classifiers, DSC—dice similarity coefficient, HD—Hausdorff Distance, GMM—gaussian mixture model, CEUS—contrast enhanced ultrasonography, PD—proton density imaging, SWE—shear wave elastography, ADC—apparent diffusing coefficient, T2W—T2 weighted imaging, T1W—T1 weighted imaging, DWI—diffusion weighted imaging, DCE—dynamic enhanced contrast imaging, SVM—support vector machine, RF—random forest, DT—decision tree, KNN—K-nearest neighbor.

**Table 4 diagnostics-11-00354-t004:** AI and machine learning (ML) techniques used to identify alterations in human genome correlating in prostate cancer (PCa), usefulness radiotherapy and in prostate cancer surgery.

Genomics Article/Reference	Genomics Analized/Image Feature	AI Method	Performance
Bertoli et al. [69]	miRNA	SVM	Diagnostic 29 miRNA AUC 0.989 ± 0.016 Prognostic signatures 7 miRNA best AUC 74.7% (CI 95%): 73.28–76.11
Karnes et al. [71]	mRNA expression, 22 genes	RF	AUC 0.79, 5 years metastasis free survival after surgery
Lee et al. [72]	Single nucleotide polymorphisms	Preconditioned random forest regression	AUC 0.70 (CI95%, 0.54–0.86, *p* = 0.01)
Hou et al. [73]	Gene expression	GA-AAN	AUC 0.953 for diagnosis AUC 0.808 for prognosis (5 year overall survival)
Liu et al. [74]	DNA methylation markers	CNN, moderated t-statistics, LASSO, and RF	CpG markers 100% sensitivity and promoter markers 92%
Hamzeh et al. [75]	Gene activity	SVM-Radial basis function kernel- SVM-RBF, Naive Bayes, RF	Highest accuracy and precision: SVM-RBF classifier Accuracy 99% HLA-DMB and EIF4G2 correlated with PCa progression
Shiradkar et al. [77]	Radiomics MRI based Multimodal co-registration scheme to map the prostate Radiomics based dose plan on MRI for brachytherapy and on CT for EBRT	Machine learning classifier- QDA	Reduction in dosage in radiomics based focal therapy compared to whole gland in EBRT and brachytherapy
Lei et al. [78]	TRUS 3D V-Net	CNN Deeply supervised V-Net	DSC 0.92 ± 0.03 HD 0.94 ± 1.55 mm MSD 0.60 ± 0.23 mm RMSD 0.90 ± 0.38 mm
Nouranian et al. [79]	TRUS, CTV, PTV	Joint sparse dictionary learning approach	Correlation between CTV and PTV= 16.28 ± 2.39%V err in estimation of PTV from CTV (single label model (TRUS/PTV) higher error vs multilable approach (*p* < 0.01)
Karimi et al. [59]	TRUS	CNN	DSC = 93.9 + /−3.5%, HD = 2.7 ± 2.3 mm (*p* = 0.01compared to other methods)
Nicolae et al. [80]	LDR treatment plan	CNN	Planning time for the ML algorithm= 0.84 ± 0.57 min compared to 17.88 ± 8.76 min for the expert planner (*p* = 0.020) Pre-implant plans were dosimetrically equivalent to the BT plans; the average prostate V150% was 4% lower for ML plans (*p* = 0.002);
Sanders et al. [81]	Seed localization performance =computing the RMSE	Sliding-window CNN algorithm	Slightly increased the run-time
**Surgery Article/Reference**	**Image Features**	**AI Method**	**Performance**
Sarikaya et al. [82]	Instrument detection and localization in robotic assisted surgery images	CNNEnd-to-end deep learning	Improves the accuracy and reduces the computation time for detection in each frame AP = 91% Training time = 7.22 h Computation time = 0.103 s, each frame
Hung et al. [83]	Automated performance data	ML algorithms	Bimanual dexterity = an ideal surgical skill

Abbreviations: RF—random forest, GA-AAN—genetic algorithm artificial neural network, RNA—Ribonucleic acid, DNA—deoxyribonucleic acid, SVM—support vector machine, AUC—area under the curve, CNN—convolutional neural network, SVM-RBF—Radial Basis Function Support Vector Machine, MRI—magnetic resonance imaging, CT—computed tomography, TRUS—transrectal ultrasonography, QDA—Ribonucleic acid, EBRT—deoxyribonucleic acid, DSC—Dice Similarity Coefficient, HD—Hausdorff distance, MSD—Mean surface distance, RMSD—Residual mean surface distance, CTV—clinical target volume, PTV—planning target volume, ML—Machine Learning, BT—Brachytherapist, RMSE—root mean square error, AP—average precision.

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
