# Peer review of "Artificial Intelligence and Machine Learning in Prostate Cancer Patient Management—Current Trends and Future Perspectives"

_diagnostics, 2021, doi:10.3390/diagnostics11020354_

Round 1
Reviewer 1 Report
Comments and Suggestions for Authors
This manuscripts reviews a topic of strong current interest and which is under rapid development: The field of prostate cancer. It takes a comprehensive approach dealing both with medical imaging and histopathology based diagnosis and different treatment modalities. This clearly makes the topic multidisciplinary, but to me the title seems somewhat misleading – the individual approaches to prostate cancer that are discussed are not necessarily multidisciplinary even though the review as a whole is, so I would suggest a different title.
It is stated that the review “focuses on the analysis of the latest and future possible advantages of AI in identifying body fluid, tissue, imaging biomarkers and treatment of prostate cancer with relevance in clinical application.” But it is unclear how this ambition has been operationalized in terms of definition of e.g. “latest” how old papers have been included? Has there been a cutoff in considering papers from before a certain year? Also consideration of whether crucial methodological papers developing AI methods on similar tissues from other organs which later have been the basis for PC tissue analysis have been considered. A more clear definition of the target domain for the review would make it easier for a reader to know if it can be expected to cover some area of his particular interest.
The paper is organized in four sections dealing with different aspects of PC: Digital pathology, diagnostic imaging, genomics, and cancer treatment, the latter divided in subsections radiotherapy and surgery. But within these sections there is very little structure to the review, just a mentioning of the different studies and some key data about the topic and findings of the studies. It would greatly improve the easy of reading the review and grasping the message if there were some added structure for instance in the form of tables summarizing studies of similar character. As an example: A systematic table of studies doing AI based Gleason grading of biopsies, TMA:s or whole sections structured according to approach for the training and amount of material and showing the achieved results and how those are compared to different ways of defining ground truth would for instance make it much easier to see how different studies relate to the current state of the art in that particular sub area. Similar structuring of other aspects would make the review as a whole more useful. As it is now its main value is the table of references, a compilation of a list of relevant recent papers in the field, not much critical analysis and structuring is offered.
The paper contains very many acronyms. Even though most of them are well known within their particular fields the broad multidisciplinary character of the paper makes it hard for a reader to follow the descriptions. Some acronyms are given explanations the first time they are used but that may be hard to find when reading. So a table where all the acronyms are given their full text explanation would be most useful.
A more detailed comment/question: At line 115 it is stated “The stratification achieved by our deep learning patch model….” Does that mean that the study in reference 19 was performed by the current authors? Please clarify.
The last paper discussed in section 2 ref 33 seems to deal with a completely different imaging approach not much related to histopathology, again an aspect of the structuring of the review. The same reference appears again in section 3 probably in a more suitable context.
The description of reference 36 on line 180 is misleading. That paper is a description of a large challenge, PROMISE-12, where many groups compared their MRI image prostate segmentation approaches on a large common dataset. The compared methods were largely based on classical image analysis but it is not correct to state that the models are outdated, some of them are the basis of more recent methods.
Also in that section more structure, e.g. a table comparing what imaging modalities have been used, what kinds of MRI sequences, TRUS, as well as methods that combine different modalities.
So in summary, the manuscript is covering an important and interesting area but could be made easier to read through more tables and more structuring of the text.
Author Response
Response to Reviewer 1 Comments
Point 1: This manuscripts reviews a topic of strong current interest and which is under rapid development: The field of prostate cancer. It takes a comprehensive approach dealing both with medical imaging and histopathology based diagnosis and different treatment modalities. This clearly makes the topic multidisciplinary, but to me the title seems somewhat misleading – the individual approaches to prostate cancer that are discussed are not necessarily multidisciplinary even though the review as a whole is, so I would suggest a different title.
Response 1: We thank the reviewer for this valuable suggestion. We changed the title accordingly. The new title is: “Artificial intelligence and machine learning in prostate cancer patient management: current trends and future perspectives”
Point 2: It is stated that the review “focuses on the analysis of the latest and future possible advantages of AI in identifying body fluid, tissue, imaging biomarkers and treatment of prostate cancer with relevance in clinical application.” But it is unclear how this ambition has been operationalized in terms of definition of e.g. “latest” how old papers have been included? Has there been a cutoff in considering papers from before a certain year? Also consideration of whether crucial methodological papers developing AI methods on similar tissues from other organs which later have been the basis for PC tissue analysis have been considered. A more clear definition of the target domain for the review would make it easier for a reader to know if it can be expected to cover some area of his particular interest.
Response 2: Thank you for this precious comment. This review focuses on the analysis of the last five years published papers on AI and ML techniques by scrutinizing PubMed, Web of Science and Science Direct, using prostate cancer, biomarkers, genomics, artificial intelligence and artificial neural networks keywords. We had also included the AI leading studies from later years or data on other diseases. The goal was to give an overview of the current evidence and future directions of AI in all the clinical management of prostate cancer patient from the diagnosis to the treatment. We added this focus at the end of the introduction section.
Point 3: The paper is organized in four sections dealing with different aspects of PC: Digital pathology, diagnostic imaging, genomics, and cancer treatment, the latter divided in subsections radiotherapy and surgery. But within these sections there is very little structure to the review, just a mentioning of the different studies and some key data about the topic and findings of the studies. It would greatly improve the easy of reading the review and grasping the message if there were some added structure for instance in the form of tables summarizing studies of similar character. As an example: A systematic table of studies doing AI based Gleason grading of biopsies, TMA:s or whole sections structured according to approach for the training and amount of material and showing the achieved results and how those are compared to different ways of defining ground truth would for instance make it much easier to see how different studies relate to the current state of the art in that particular sub area. Similar structuring of other aspects would make the review as a whole more useful. As it is now its main value is the table of references, a compilation of a list of relevant recent papers in the field, not much critical analysis and structuring is offered.
Response 3: We thank the reviewer for this valuable suggestion. We added tables accordingly.
Point 4: The paper contains very many acronyms. Even though most of them are well known within their particular fields the broad multidisciplinary character of the paper makes it hard for a reader to follow the descriptions. Some acronyms are given explanations the first time they are used but that may be hard to find when reading. So, a table where all the acronyms are given their full text explanation would be most useful.
Response 4: We thank the reviewer for this suggestion. We added a table with the acronyms accordingly.
Point 5: A more detailed comment/question: At line 115 it is stated “The stratification achieved by our deep learning patch model….” Does that mean that the study in reference 19 was performed by the current authors? Please clarify.
Response 5: We thank the reviewer for this comment. We changed the text accordingly.
Point 6: The last paper discussed in section 2 ref 33 seems to deal with a completely different imaging approach not much related to histopathology, again an aspect of the structuring of the review. The same reference appears again in section 3 probably in a more suitable context.
Response 6: We appreciate this comment and changed the reference accordingly.
Point 7: The description of reference 36 on line 180 is misleading. That paper is a description of a large challenge, PROMISE-12, where many groups compared their MRI image prostate segmentation approaches on a large common dataset. The compared methods were largely based on classical image analysis but it is not correct to state that the models are outdated, some of them are the basis of more recent methods.
Response 7: We greatly appreciate this suggestion and changed the text accordingly.
Point 8: Also in that section more structure, e.g. a table comparing what imaging modalities have been used, what kinds of MRI sequences, TRUS, as well as methods that combine different modalities.
Response 8: We thank the reviewer for this comment. We added the table accordingly.
Point 9: So in summary, the manuscript is covering an important and interesting area but could be made easier to read through more tables and more structuring of the text.
Response 9: We are thankful to the reviewer for this evaluation. Accordingly, we added more tables and we modified the structure of the text, adding different paragraphs for each topic.
Reviewer 2 Report
Comments and Suggestions for Authors
This manuscript reviews the use of AI and machine learning (ML) in the diagnosis and management of prostate cancer. This is a rapidly growing area and one of great interest. Overall, authors summarize a lot of data, but literature is listed in one long paragraph for each topic, and interpretation for reader about limitations and whether work is ready for routine clinical use is for the most part lacking. The abstract, introduction, and conclusion lack critical judgement and mislead reader to conclude that AI is being widely utilized in routine clinical practice. Additional items listed below:
Abstract: explain "deep learning"
Big claim that AI can predict toxicity - literature suggesting this is still sparse - I recommend backing off on tone of this claim.
Intro: AI and ML is very promising, but as of yet not widely applied or ready for "primetime". Please provide more context/tone to note that this is still an area of development
Pathology/Imaging: nice review. Please provide some summary - together, are these studies sufficient to show the necessity of using AI at this time? Are any of these methods approved by government agency (comments on whether such such approval is required?) Finally, please note that one main benefit of AI may be the ability to provide quality control across experience and training levels
I disagree with calling genomic classifiers AI. Perhaps the generation of these classifiers used computer algorithms to find the most predictive model but clinical use of classifiers uses a set calculation and does not include machine learning or an adaptive decision-making process.
Author Response

(The authors gave the same response as above.)

Round 2
Reviewer 2 Report
Comments and Suggestions for Authors
This review has been significantly edited and organized, and to the authors' credit provides a much more robust and straightforward review in its current state. A few minor suggesitons/clarifications.
2.1.4 and 2.1.5 seem to be the same topic and could perhaps be combined.
In 5.1.1, could you provide more detail as to whether the application of ML MRI findings to treatment planning had any effect (planning changes, toxicity, cancer control, etc)? Otherwise this section is quite sparse.
Author Response
Point 1: 2.1.4 and 2.1.5 seem to be the same topic and could perhaps be combined.
Response 1: We thank the reviewer for this suggestion. We modified the structure of the text, by combining the two paragraphs and modified the head title of the paragraph.
At line: 160
Modified in “2.1.4 Whole slide image analysis in prostate biopsies (cancer detection and grading) and performance comparison of AI models and pathologists”
We have deleted the headline at line number: 203.
Point 2: In 5.1.1, could you provide more detail as to whether the application of ML MRI findings to treatment planning had any effect (planning changes, toxicity, cancer control, etc)? Otherwise this section is quite sparse.
Response 2: We thank the reviewer for this suggestion. Accordingly we added more studies that evaluate the impact of MRI in treatment planning, dose evaluation and toxicity.
From line 488